# First Isolation of Methicillin-Resistant Livestock-Associated *Staphylococcus aureus* CC398 and CC1 in Intensive Pig Production Farms in Argentina

**DOI:** 10.3390/ani13111796

**Published:** 2023-05-29

**Authors:** Paula Gagetti, Gabriela Isabel Giacoboni, Hernan Dario Nievas, Victorio Fabio Nievas, Fabiana Alicia Moredo, Alejandra Corso

**Affiliations:** 1Servicio Antimicrobianos, INEI-ANLIS “Dr. Carlos G. Malbrán”, Laboratorio Nacional/Regional de Referencia en Resistencia a los Antimicrobianos PAHO, Buenos Aires 1281, Argentina; 2Laboratorio de Bacteriología y Antimicrobianos, Facultad de Ciencias Veterinarias, Universidad Nacional de La Plata, Buenos Aires 1900, Argentina; gigiacoboni@gmail.com (G.I.G.); hernandnievas@gmail.com (H.D.N.); vfnievas@yahoo.com.ar (V.F.N.); famoredo@yahoo.com.ar (F.A.M.)

**Keywords:** livestock-associated methicillin-resistant *Staphylococcus aureus* (LA-MRSA), pigs, ST398, whole-genome sequencing, zoonosis

## Abstract

**Simple Summary:**

Methicillin-resistant *Staphylococcus aureus* (MRSA) is an important pathogen that causes healthcare- and community-associated infections in humans. Livestock-associated methicillin-resistant *Staphylococcus aureus* (LA-MRSA) has emerged and has been disseminated among pigs worldwide. In the present work, the first LA-MRSA isolated from nasal colonization in healthy fattening pigs in Argentina was characterized. The isolates showed a high degree of multi-drug resistance, and the genomic characterization revealed the presence of relevant resistance genes in the isolates. The present study revealed that LA-MRSA colonizing healthy pigs in Argentina belongs to CC398 and CC1, two MRSA lineages frequently associated to pigs in other countries.

**Abstract:**

Since the mid-2000s, livestock-associated methicillin-resistant *Staphylococcus aureus* (LA-MRSA) has been identified among pigs worldwide, CC398 being the most relevant LA-MRSA clone. In the present work, nasal swabs were taken from healthy pigs of different age categories (25 to 154 days) from 2019 to 2021 in four intensive farms located in three provinces of Argentina. The aim of the present study was to characterize the first LA-MRSA isolates that colonized healthy fattening pigs in Argentina in terms of their resistance phenotype and genotype and to know the circulating clones in the country. Antimicrobial susceptibility, presence of the *mecA* gene and PCR screening of CC398 were evaluated in all the isolates. They were resistant to cefoxitin, penicillin, tetracycline, chloramphenicol and ciprofloxacin but susceptible to nitrofurantoin, rifampicin, vancomycin and linezolid. Furthermore, 79% were resistant to clindamycin and lincomycin, 68% to erythromycin, 58% to gentamicin and 37% to trimethoprim/sulfamethoxazole. All the isolates were multidrug resistant. The clonal relation was assessed by *SmaI*-PFGE (pulsed-field gel electrophoresis) and a representative isolate of each PFGE type was whole genome sequenced by Illumina. MLST (multilocus sequence typing), resistance and virulence genes and SCC*mec* typing were performed on sequenced isolates. The isolates were differentiated in three clonal types by PFGE, and they belonged to sequence-type ST398 (58%) and ST9, CC1 (42%) by MLST. SCC*mec* typeV and several resistance genes detected showed complete correlation with resistance phenotypes. The present study revealed that LA-MRSA colonizing healthy pigs in Argentina belongs to CC398 and CC1, two MRSA lineages frequently associated to pigs in other countries.

## 1. Introduction

*Staphylococcus aureus* is a commensal microorganism of the skin and nose of humans and many animals, but it is also an important human and animal opportunistic pathogen that may be involved in a wide variety of infections, from mild to severe [1]. *S. aureus* has the capacity to acquire a wide diversity of antimicrobial resistance and virulence genes, and particularly, methicillin-resistant *S. aureus* (MRSA) has relevance to human and animal health.

Methicillin-resistant *Staphylococcus aureus* (MRSA) can be classified into three epidemiological groups according to its most frequent niche and associated clonal complexes. These three groups are Healthcare-Associated (HA-MRSA), Community-Associated (CA-MRSA) and Livestock-Associated (LA-MRSA) [2].

Since the 2000s, LA-MRSA emerged and spread worldwide in pigs [3]. The first report of MRSA in pigs was from the Netherlands in 2005 [4]. In the following years, an increasing number of clonal complex CC398 MRSA isolates, initially named pig-associated MRSA and later designated as LA-MRSA, was revealed [1].

Although in European countries MRSA was recovered from pigs mainly belonging to CC398 [3,5], other minor clones associated with pig colonization were also found including ST1, ST9 and ST97 [6]. CC398 MRSA has also been detected in pigs in North America, South America and Australia [7,8,9], and it has been occasionally observed in Asia, where CC9 is the major LA-MRSA [10]. Pigs are often colonized by MRSA ST398, but infection by this species is rare [6].

The presence of MRSA CC398 in animals constitutes a risk due to the possible transfer of these strains to humans. Several studies have identified high rates of MRSA CC398 colonization in human subjects occupationally exposed to animals, such as owners, farmers and veterinarians. However, human-to-human transmission of MRSA CC398 occurs rarely compared to other MRSA lineages. In a study conducted in 17 countries in Europe, the proportion of MRSA ST398 isolates from humans represented less than 2% of all MRSA isolates [11].

The use of high levels of antibiotic in agriculture combined with intensive farming has raised concerns about livestock as a reservoir of antibiotic-resistant human infections [12].

Until the present study, investigations on LA-MRSA in Argentina were limited. In a previous study, we evaluated the nasal colonization of MRSA in fattening pigs from four farms located in three Argentinian provinces, Buenos Aires, Santa Fe and San Luis. Between 2019 and 2021, a total of 64 piglets aged 25–154 days were screened for MRSA colonization, and 19 LA-MRSA were isolated [13]. Herein, we carry out the characterization of the recovered isolates to investigate their genomic relatedness.

The aim of the present study was to characterize the first LA-MRSA isolated from pigs in Argentina using distinct molecular techniques and to determine their clonal diversity.

## 2. Materials and Methods

### 2.1. Sampling and MRSA Identification

Samples were taken from healthy pigs aged between 25 and 154 days in four intensive breeding farms located in three provinces of Argentina. Between June and July 2019, 42 piglets aged 25, 77, 100, 110 and 154 days from a farm located in the Buenos Aires province (Farm 1) were swabbed. During July 2021, 22 piglets were swabbed, 8 27-day-old piglets from a farm in the Santa Fe province (Farm 2), 8 80-day-old piglets from a farm in the Buenos Aires province (Farm 3) and 6 120-day-old piglets from a farm in the San Luis province (Farm 4). Samples were taken by swabbing a single nasal cavity of each animal with a 15 cm long sterile cotton swab from the deep area of the ventral turbinate. The swabs were inoculated into an enrichment tryptic soy broth containing 6.5% NaCl, incubated overnight at 35 °C and seeded onto chromogenic agar CHROMagarTM MRSA (CHROMagar, Paris, France). Plates were incubated at 35 °C for 24 h, and suspicious colonies were confirmed as S. aureus by conventional biochemical tests.

### 2.2. Susceptibility Testing

Antimicrobial susceptibility tests were performed by disk diffusion and/or Vitek 2 (BioMerieux, Marcy-l’Étoile, France) according to CLSI for the following antibiotics (disk concentration in brackets): cefoxitin (30 µg), penicillin (10 units), gentamicin (10 µg), erythromycin (15 µg), clindamycin (2 µg), tetracycline (30 µg), chloramphenicol (30 µg), trimethoprim/sulfamethoxazole (1.25/23.75 µg), rifampicin (5 µg), ciprofloxacin (5 µg), nitrofurantoin (300 µg), linezolid (30 µg) and vancomycin MIC. All antimicrobial susceptibility tests were interpreted according to CLSI guidelines [14]. Multidrug-resistant (MDR) isolates were defined for those isolates showing phenotypic resistance to at least three antimicrobials from different classes [15].

### 2.3. Gene Detection by PCR

PCR was performed on all LA-MRSA strains to detect the presence of the *mecA* and a specific sequence of *sau1-hsdS1* gene, markers for methicillin resistance, and CC398, respectively, as previously described [16,17]. *S. aureus* ATCC 43300 and *S. aureus* ATCC 29213 were used as *mecA* positive and negative controls, respectively.

### 2.4. Molecular Typing

The 19 isolates were characterized by pulsed-field gel electrophoresis (PFGE) of total DNA restricted with the enzyme SmaI (New England Biolabs, Beverly, MA, USA) as previously described [18]. PFGE (pulsed-field gel electrophoresis) was performed by clamped homogeneous electric field electrophoresis with a CHEF DR III System (Bio-Rad Laboratories, Richmond, CA, USA) under the following conditions: switch time, 2.0 to 20.0 s and running time, 20 h; temperature 11.3 °C, angle 120° and voltage 6 V/cm. Separated DNA fragments were stained with ethidium bromide and visualized with a UV transilluminator. Banding patterns were analysed by visual inspection and interpreted according to Tenover criteria [19].

MLST (multilocus sequence typing) was carried out by MLST 2.0 [20] available at the Center for Genomic Epidemiology (http://www.genomicepidemiology.org/, accessed on 20 December 2022). To assign sequence types (STs) and clonal complexes (CCs), the pubMLST *S. aureus* multilocus sequence typing database was used [21].

### 2.5. Genome Sequencing and Analysis

Four selected isolates were whole genome sequenced (WGS) according to their PFGE profile. Whole bacterial DNA was extracted with QIAcube, using the QIAamp1 DNA Mini Kit (Qiagen, Valencia, CA, USA), and sequenced at the Genomic and Bioinformatic Platform, INEI-ANLIS “Dr. Carlos G. Malbrán” using Nextera XT DNA Sample Prep Kit (Illumina, San Diego, CA, USA) on a MiSeq sequencer to generate 250 bp paired end reads approach. De novo assembly and annotation were performed using PATRIC (Pathosystems Resource Integration Center) software CLI Release 1.039 (https://www.bv-brc.org/, accessed on 20 December 2022). Detection of resistance genes was carried out by ResFinder v4.1 [22] and PATRIC using the available CARD (Comprehensive Antimicrobial Resistance Database) and NDARO (National Database of Antibiotic Resistant Organisms) databases. The antimicrobial resistance genes content was compared with the phenotype presented by the isolates.

SCC*mec*Finder 1.2 from the Center for Genomic Epidemiology (CGE) database was used to determine the Staphylococcal chromosome cassette *mec* (SCC*mec*) type (available at https://cge.cbs.dtu.dk/services/SCCmecFinder/, accessed on 22 November 2022).

Virulence genes were detected using VirulenceFinder v2.0 [23], (available at https://cge.food.dtu.dk/services/VirulenceFinder/, accessed on 20 December 2022).

The plasmid replicons were analysed with PlasmidFinder 2.0 [24] (available at https://cge.food.dtu.dk/services/PlasmidFinder/, accessed on 22 December 2022), and the presence of mobile genetic elements and their relation to antimicrobial resistance genes were predicted with MobileFinder 1.0.3 [25] (available at https://cge.food.dtu.dk/services/MobileElementFinder/, accessed on 24 December 2022).

PATRIC software was used to generate a single-nucleotide polymorphism (SNP) alignment comparing core-genes, and this comparison was used to construct the phylogenetic tree using the RAxML Fast Bootstrapping method. *S. aureus* strain ISU926 isolate ST398 (accession number CP017091.1) was used as a reference genome.

### 2.6. Data Availability

Sequencing data have been submitted to the NCBI Sequence Read Archive (SRA) database under the BioProject accession number PRJNA936104.

## 3. Results

In total, 19 MRSA were isolated from the 64 swabs processed (29.7%). They were recovered from piglets of 27, 80, 120 and 154 days of age as follows: 2/42 of 154 days on Farm 1, 4/8 of 27 days on Farm 2, 7/8 of 80 days on Farm 3 and 6/6 of 120 days of age in Farm 4.

The 19 isolates showed resistance to penicillin, cefoxitin, tetracycline, chloramphenicol and ciprofloxacin. Resistance to clindamycin and lincomycin was 79%, to erythromycin was 68%, to gentamicin was 58% and to trimethoprim/sulfamethoxazole was 37%. None of the isolates presented resistance to rifampicin, nitrofurantoin, vancomycin and linezolid.

All the isolates were MDR, being resistant between five and seven different antimicrobial families and displaying five different profiles. The most frequent profile, i.e., resistance to β-lactams, tetracyclines, phenicols, fluoroquinolones and macrolides/lincosamides was found in six isolates. The second resistance profile including β-lactams, tetracyclines, phenicols, fluoroquinolones, macrolides/lincosamides, aminoglycosides and trimethoprim/sulfamethoxazole was displayed by five isolates. The eight remaining isolates showed three different resistance profiles as shown in Table 1.

Using SmaI-PFGE, 12/19 LA-MRSA isolates were differentiated in 3 clonal types: A (n: 2), B (n: 4) and C (n: 6). The seven remaining isolates were repeatedly untypeable by SmaI-PFGE. Using MLST, clone B and untypeable isolates belonged to ST398, CC398 (11/19; 58%). Clone A and C isolates belonged to ST9, CC1 (8/19; 42%).

The *mecA* gene was detected by PCR in the 19 LA-MRSA isolates, and the specific sequence of the *sau1-hsdS1* gene characteristic of CC398 was only detected in 11/19 isolates, in concordance with the PFGE and MLST results.

WGS was performed to four representative isolates, selected based on their PFGE clonal type. Resistome analysis corroborated the presence of several genes in the genome, consistent with the antimicrobial resistance profiles of the isolates. All sequenced isolates carried *mecA* gene encoding PBP2a and the β-lactamase gene *blaZ*, the phenicol exporter *fexA* gene and *tet*(*M*), *tet*(*L*), *tet*(*K*) and/or *tet38* genes encoding tetracycline resistance (Figure 1).

Phenotypic resistance to ciprofloxacin was displayed by all the isolates, and the four sequenced isolates carried mutations in ParC (S80Y in three of them and S80F in the other one) and GyrA (S84L). Gentamicin-resistant isolates harboured the *aac*(*6*′)*-Ie-aph*(*2*″)*-Ia*, *ant*(*6*)*-I* gene encoding an aminoglycoside acetyltransferase. Erythromycin-resistant isolates carried the *ermC* gene, and isolates resistant to lincosamide carried the *lnuB*, *lsa*(*E*) or *vgaA* genes. Isolates resistant to trimethoprim/sulfamethoxazole carried the dihydrofolate reductase encoding *dfrG* gene (Figure 1).

As displayed in Figure 1, we found complete concordance between phenotype and genotype for all the antibiotic tested except for the isolate M8743, which presented susceptibility to trimethoprim/sulfamethoxazole despite carrying the *dfrG* gene.

All the sequenced isolates carried *mecC2* complex and *ccrC1* and were classified as SCC*mec* type V (5C2&5).

Four virulence factors (*aur*, *hlgA*, *hlgB* and *hlgC*) were present in all the sequenced isolates. Additionally, egc cluster (*seg*, *sei*, *sem*, *sen*, *seo* and *seu* genes) encoding enterotoxins were present only in M8695, a CC1 isolate.

Neither Panton–Valentine leucocidin genes *lukF* and *lukS* nor genes belonging to the human immune evasion cluster, *scn* and *sak*, were found in any of the strains.

The plasmid replicon sequences rep10 and rep7b were identified in M8695 carrying *ermC* and *vgaA* genes, respectively. In M8743 rep7a, rep10 carrying *ermC* gene, rep22 carrying *aaD* gene and repUS43 carrying *tet*(*M*) gene were present. Plasmid replicons rep10 carrying *ermC*; rep13, rep22 carrying *aaD*; and repUS43 carrying *tet*(*M*) gene were identified in M8746, and finally, rep22 carrying *aaD* and repUS28 carrying *tet*(*L*) were detected in M8773.

The *fexA* gene was carried by MGE (Mobile Genetic Elements) Tn*558* in the four sequenced isolates. Additionally, other MGE were detected: IS*Sau3* (IS*1182*) in M8695, IS*Sau1* (IS*30*) in M8743 and M8746, IS*Sau5* (IS*30*) in M8746 and IS*256* in M8746 and M8773 isolates.

In agreement, the phylogenetic tree shows that isolates M8743 and M8746 clustered with the reference strain ISU926 ST398, unlike ST9 isolates M8695 and M8773 (Figure 2).

As shown in Table 2, the composition of LA-MRSA genomes was analysed with PATRIC software. They showed an average size of 2,606,243 bp, with an average 2478 genes annotated (range 1928 to 2673).

## 4. Discussion

The increase in antimicrobial resistance in MRSA isolates from pigs in recent years has been considered a matter of great concern [3,26]. Selective pressure due to antibiotic misuse/overuse in animal fattening units leads to the selection of a subpopulation of MRSA that, in addition to resistance to methicillin, presents cross-resistance to several other molecules [27].

Multidrug resistance was observed in all the LA-MRSA isolates studied, and five different resistance patterns were found, with resistance to beta-lactams, tetracyclines, chloramphenicol, fluoroquinolones, aminoglycosides, lincosamides, macrolides and trimethoprim/sulfamethoxazole. This is consistent with recent studies reporting high level of multi-drug resistance in LA-MRSA isolated from pigs [28], which originated from the acquisition of genes encoding resistance to several antibiotic families by horizontal gene transfer from other staphylococci and bacteria of human and animal origin [29]. Macrolides, lincosamides, β-lactams, tetracyclines and sulfonamides are frequently used in pig production as curative, prophylactic or metaphylactic treatments [30].

The resistance profiles detected in LA-MRSA were notably different from those detected in MRSA isolated from humans in the same period of time, which, although most of them were susceptible to nitrofurantoin, rifampicin, vancomycin and linezolid, presented less than 30% of resistance to erythromycin, clindamycin and gentamicin, 12% to ciprofloxacin and less than 5% to tetracycline and trimethoprim/sulfamethoxazole [31].

The different resistance profiles observed in MRSA of human and porcine origin are mainly due to the different use of antimicrobials in human health and in food-producing animals and may also be related to the massive use of antibiotics in livestock.

MRSA strains can be transmitted between different animal species and to humans who come in close contact with colonized animals, such as veterinarians and farm workers. On the other hand, colonized humans can also transmit MRSA to other humans and between animal settings. Wherever there is an interface between different host species, there is the possibility of bacterial exchange [1]. Nonetheless, pigs are considered important hosts for zoonotic transmission of S. aureus to humans [2].

Regarding CC398 lineage, recent studies of phylogenetic analysis showed that strains of human, animal and food origin cluster in different clades and have specific genetic markers. The presence of the phage φ3 is related to human adaptation, and the avian prophage φAvβ is a genetic marker for avian adaptation. The most relevant aspect of the CC398 lineage that must be highlighted is the plasticity of its genome, which allows it to acquire genetic elements related to adaptation to different hosts, antimicrobial resistance and virulence [1]. It should be noted that the colonization of pigs with these MDR strains constitutes a reservoir of resistance genes capable of spreading by mobile genetic elements to bacteria that colonize other hosts.

Although SmaI is the restriction enzyme of choice for *S. aureus* fingerprinting by PFGE, seven isolates were untypeable by SmaI-PFGE but were typed as ST398 by MLST. Restriction failures particularly associated with hospital-acquired and LA-MRSA ST398 strains isolates were previously reported [32].

To detect *S. aureus* CC398 strains, the PCR developed by Stegger et al. [17] was used. This PCR is based on the differences in the sequence of the *sau1-hsdS1* gene responsible for the restriction modification specificity of this bacterial species and allowed us to categorize isolates from pulsotype B and the nontypable isolate3 as CC398. This specific PCR represents a very useful tool for the screening of LA-MRSA CC398 isolates in a very simple way and at a very low cost.

Antimicrobial resistance gene patterns were typical for strains isolated from pigs [28], and there was little variation between sequenced isolates. All of them carried *mecA*, *blaZ*, *fexA* and at least one gene encoding tetracycline resistance. The isolates also harboured *tet*(*M*), *tet*(*L*), *tet*(*K*) and/or *tet38* genes. Tetracycline resistance mediated by the presence of *tet*(*M*) gene alone or in combination with other *tet* genes is a marker for the detection of LA-MRSA CC398 [33,34,35] as was observed in the present study. In MRSA CC398 strains, the *tet*(*M*) gene is integrated into the SCC*mec* element [36]. Such high levels of tetracycline resistance could be consequence of the intensive use of this antibiotic in agriculture, which may favour the selection resistant bacteria [27].

Some strains harboured resistance genes *ermC*, *lnuB* and *lsa*(*E*), conveying resistance to macrolides, lincosamides and streptogramin B, the *aac*(*6*′)*-Ie-aph*(*2*″)*-Ia*, *ant*(*6*)*-I*, *ant*(*9*)*-Ia* and/or *aadD* genes conferring resistance to aminoglycosides and *dfrG* gene conferring resistance to trimethoprim/sulfamethoxazole. Mutations in ParC and GyrA associated to fluoroquinolone resistance detected in the sequenced isolates were previously reported [37].

Complete correlation between phenotype and genotype was observed for all the antibiotic tested except the isolate M8743, which presented susceptibility to trimethoprim/sulfamethoxazole despite carrying the *dfrG* gene. Comparison between nucleotide sequences of the *dfrG* gene from isolate M8743 susceptible to SXT and M8746 resistant to SXT showed that both were identical. Therefore, we postulate that the phenotype discrepancies observed may be linked to gene expression.

All the strains harboured SCC*mec* type V (5C2&5), the dominant SCC*mec* sub-type in LA-MRSA CC398 from several European countries [38,39].

Regarding the virulence factors, the aur and *hlg* genes encoding aureolysin and gamma-hemolysin, respectively, were carried by the four sequenced isolates. The enterotoxin cluster *egc* was only present in one CC1 isolate. The *egc* cluster consists of six genes (*seg*, *sei*, *sem*, *sen*, *seo* and *seu*), encoding enterotoxins probably involved in staphylococcal food poisoning outbreaks that are part of the *S. aureus vSaβ* genomic island [40]. All the isolates studied in the present work were PVL-negative, which is typical for the LA-MRSA CC398 clade [38]. The same was observed in CC1 isolates from animals, whereas human isolates were generally PVL-positive [1,33].

Due to the limitations of short-read sequencing technologies, we were unable to determine the location of some resistance genes on the plasmids. However, plasmid replicons associated with some resistance genes were found, such as the repUS43 replicon that carried the *tet*(*M*) gene detected in both CC398 isolates. We also could predict that the *fexA* gene was carried by MGE Tn*558* in the four sequenced isolates.

LA-MRSA CC398 strains have been described in many countries worldwide, with different prevalence in pork livestock [41]. Although CC398 is the main lineage associated with LA-MRSA, other clonal complexes and sequence types which are not within CC398 have also been associated with livestock. In the present study, 19 LA-MRSA were isolated from nasal colonization of healthy piglets of different age categories in four intensive pig production farms located in three provinces of Argentina. It is noteworthy that ST398 isolates were recovered from 11 pigs in two farms from the Buenos Aires and Santa Fe provinces. The remaining eight isolates recovered in the Buenos Aires and San Luis provinces belonged to CC1, a successful lineage associated with human infections, which includes PVL-positive CA-MRSA ST1 also known as USA400 [42]. In Argentina, most CA-MRSA from human infections belonged to two main clones: ST30-SCC*mec*IVc-PVL+ and ST5-IV-SCC*mec*IVa-PVL+. CA-MRSA belonging to CC1 were detected in a very low proportion, but they were ST1 not ST9 [43]. However, CC1 SCC*mec*V is one of the major livestock-associated lineages in the pig farming industry in Italy and is associated with pigs in other European countries [44].

## 5. Conclusions

In summary, LA-MRSA from Argentinian fattening pigs was represented by two genetic lineages, CC398 and CC1, frequently associated with pigs in other countries. The present study alerts about the emergence of CC398 in Argentina and underscores the importance of monitoring the evolution of LA-MRSA in pig farms in order to implement control measures and reduce the risk of spread in animal and human populations.

## Figures and Tables

**Figure 1 animals-13-01796-f001:**
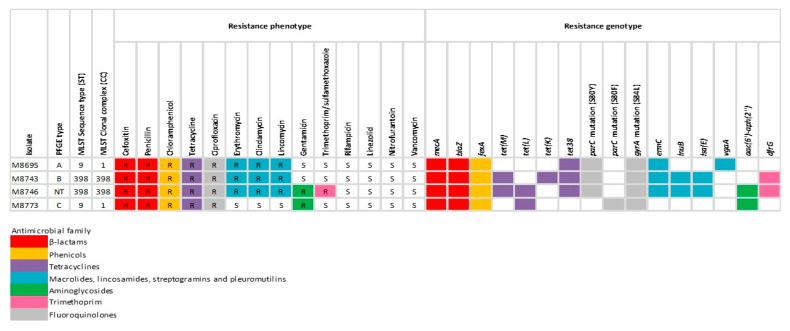
Resistance phenotype versus genotype obtained by whole genome sequencing of four LA-MRSA representative isolates from pigs. Coloured squares indicate presence of gene. PFGE: pulsed-field gel electrophoresis, MLST: multilocus sequence typing, CC: clonal complex, ST: sequence type, R: resistant, S: susceptible.

**Figure 2 animals-13-01796-f002:**
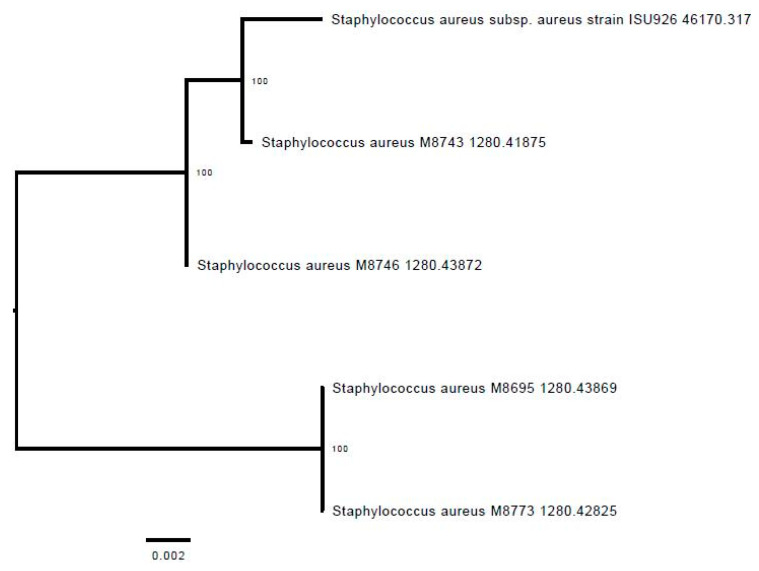
Phylogenetic tree of the four sequenced LA-MRSA compared with a CC398 reference strain ISU925.

**Table 1 animals-13-01796-t001:** Resistance phenotype and multidrug resistance patterns of LA-MRSA (n: 19).

No. of Isolates	Resistance Phenotype	No. of Antimicrobial Classes
5	FOX, PEN, TET, CMP, CIP, ERY, CLI, LIN, GEN, SXT	7
2	FOX, PEN, TET, CMP, CIP, CLI, LIN, SXT	6
2	FOX, PEN, TET, CMP, CIP, ERY, CLI, LIN, GEN	6
6	FOX, PEN, TET, CMP, CIP, ERY, CLI, LIN	5
4	FOX, PEN, TET, CMP, CIP, GEN	5

FOX: cefoxitin, PEN: penicillin, TET: tetracycline, CMP: chloramphenicol, ERY: erythromycin, CLI: clindamycin, LIN: lincomycin, CIP: ciprofloxacin, GEN: gentamicin, SXT: trimethoprim/sulfamethoxazole.

**Table 2 animals-13-01796-t002:** Whole Genome Characterization of four LA-MRSA sequenced (n: 4).

Features	M8695	M8743	M8746	M8773
Farm region	Buenos Aires	Santa Fe	Buenos Aires	San Luis
Genome ID	128.043.869	128.041.875	128.043.872	128.042.825
Contigs	74	64	126	54
Genome length (bp)	2,781,818	2,783,192	2,794,082	2,065,882
GC Content (%)	32.64	32.80	32.73	33.11
MLST	ST1 (CC9)	ST398 (CC398)	ST398 (CC398)	ST1 (CC9)
Resistance genes	*mecA*; *blaZ*; *fexA*; *tet3*8; *ermC*; *vgaA*; *ant*(*6*)*-I*; *ant*(*9*)*-Ia*	*mecA*; *blaZ*; *fexA*; *tetM*; *tetK*; *tet38*; *ermC*; *lnuB*; *lsa*(*E*); *aadD*; *dfrG*	*mecA*; *blaZ*; *fexA*; *tetM*; *tetL*; *tet38*; *ermC*; *lnuB*; *lsa*(*E*); *aac*(*6*′)*-aph*(*2*″); *aadD*; *dfrG*	*mecA*; *blaZ*; *fexA*; *tetL*; *aac*(*6*′)*-aph*(*2*″); *aadD*
SCC*mec* elements	type_V(5C2&5)	type_V(5C2&5)	type_V(5C2&5)	type_V(5C2&5)
Virulence genes	*aur*, *hlgA*, *hlgB*, *hlgC*, *seg*, *sei*, *sem*, *sen*, *seo*, *seu*	*aur*, *hlgA*, *hlgB*, *hlgC*	*aur*, *hlgA*, *hlgB*, *hlgC*	*aur*, *hlgA*, *hlgB*, *hlgC*
Plasmid replicons	rep10, rep7b	rep7a, rep10, rep22, repUS43	rep10, rep13, rep22, repUS43	rep22, repUS28
MGE *	Tn*558*, IS*Sau3* (IS*1182*)	Tn*558*, IS*Sau1* (IS*30*)	Tn*6009*, Tn*558*, IS*256*, IS*Sau1* (IS*30*)	Tn*558*, IS*Sau5* (IS*30*), IS*256*

* Mobile Genetic Elements.

## Data Availability

The sequences data generated in this study have been deposited in the GenBank under BioProject PRJNA936104.

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
