# Peer review of "First Isolation of Methicillin-Resistant Livestock-Associated Staphylococcus aureus CC398 and CC1 in Intensive Pig Production Farms in Argentina"

_animals, 2023, doi:10.3390/ani13111796_

Round 1
Reviewer 1 Report
The introduction section is fine.
The summary must be modified in the aspects that I expose in the other sections, in methods, results and discussion.
The methods section is very incomplete, since it does not explain the origin of the animals, it indicates mass production farms, but not where the samples were collected, whether farms or slaughterhouses, at what time, how many samples, etc...... ………How were the animals selected?
I believe that all these data have to be well explained, as well as indicate what type of study it is, date, etc...
I think this section should be rewritten by adding all this data and also adding it in the results and discussing it in the discussion section.
The results are also incomplete. There are no total prevalence values, nor by farms, in accordance with the material and methods section, this must also be well described. It is important to know these isolates, which indicates (19) from which total number of samples they come.
Were other non-aureus multiresistant Staphylococcus species determined? There are several studies that indicate the increasing prevalence of these strains in samples of animal origin, especially in pigs.
Discussion
The same sections that do not appear in the previous sections must be discussed. If these 19 isolates have been found from healthy animals, on farms or in slaughterhouses, etc. and compare with other studies.
Conclusions section is fine
Reviewer 2 Report
1.Line 214-216: These conclusive descriptions have already appeared in Results, it is recommended to briefly mention them or alter the description.
2.Line 226: After comparing the pig derived strains with human derived strains, it is suggested to further discuss and analyze the pros and cons of these differences to livestock production and human life.
3.Line 236: This paragraph can be briefly described and should be followed by analyses and discussion of the PCR results in this experiment
Totally speaking, the content of the experiment was limited, the analysis of the results was not thorough enough, and other relevant properties of the newly isolated strains was rarely explored. The discussion was comprehensive and rich in content, however, the drawback was that content related to production and daily life was rarely mentioned, such as the impact of the isolated bacterium on animal husbandry and human health.
1.Line 214-216: These conclusive descriptions have already appeared in Results, it is recommended to briefly mention them or alter the description.
2.Line 226: After comparing the pig derived strains with human derived strains, it is suggested to further discuss and analyze the pros and cons of these differences to livestock production and human life.
3.Line 236: This paragraph can be briefly described and should be followed by analyses and discussion of the PCR results in this experiment
Totally speaking, the content of the experiment was limited, the analysis of the results was not thorough enough, and other relevant properties of the newly isolated strains was rarely explored. The discussion was comprehensive and rich in content, however, the drawback was that content related to production and daily life was rarely mentioned, such as the impact of the isolated bacterium on animal husbandry and human health.
Round 2
Reviewer 1 Report
I believe that with the modifications made, the work can be published in the journal